# Effects of Two Randomized and Controlled Multi-Component Interventions Focusing On 24-Hour Movement Behavior among Office Workers: A Compositional Data Analysis

**DOI:** 10.3390/ijerph18084191

**Published:** 2021-04-15

**Authors:** Lisa-Marie Larisch, Emil Bojsen-Møller, Carla F. J. Nooijen, Victoria Blom, Maria Ekblom, Örjan Ekblom, Daniel Arvidsson, Jonatan Fridolfsson, David M. Hallman, Svend Erik Mathiassen, Rui Wang, Lena V. Kallings

**Affiliations:** 1Department of Physical Activity and Health, The Swedish School of Sport and Health Sciences (GIH), 114 86 Stockholm, Sweden; lisa-marie.larisch@gih.se (L.-M.L.); carla.nooijen@gih.se (C.F.J.N.); victoria.blom@gih.se (V.B.); maria.ekblom@gih.se (M.E.); orjan.ekblom@gih.se (Ö.E.); rui.wang@gih.se (R.W.); lena.kallings@gih.se (L.V.K.); 2Division of Insurance medicine, Department of Clinical Neuroscience, Karolinska Institutet, 171 77 Stockholm, Sweden; 3Department of Neuroscience, Karolinska Institutet, 171 77 Stockholm, Sweden; 4Center for Health and Performance, Department of Food and Nutrition and Sport Science, Faculty of Education, University of Gothenburg, 405 30 Gothenburg, Sweden; daniel.arvidsson@gu.se (D.A.); jonatan.fridolfsson@gu.se (J.F.); 5Centre for Musculoskeletal Research, Department of Occupational Health Sciences and Psychology, University of Gävle, 801 76 Gävle, Sweden; David.hallman@hig.se (D.M.H.); Svenderik.mathiassen@hig.se (S.E.M.); 6Division of Clinical geriatrics, Department of Neurobiology, Care Sciences and Society (NVS), Karolinska Institutet, 141 52 Huddinge, Sweden; 7Wisconsin Alzheimer’s Disease Research Center, University of Wisconsin School of Medicine and Public Health, Madison, WI 53792, USA; 8Department of Public Health and Caring Sciences, Family Medicine, Uppsala University, 751 22 Uppsala, Sweden

**Keywords:** physical activity, sedentary behavior, 24-h movement behavior, compositional data analysis, office workers, cognitive behavioral therapy

## Abstract

Intervention studies aiming at changing movement behavior have usually not accounted for the compositional nature of time-use data. Compositional data analysis (CoDA) has been suggested as a useful strategy for analyzing such data. The aim of this study was to examine the effects of two multi-component interventions on 24-h movement behavior (using CoDA) and on cardiorespiratory fitness among office workers; one focusing on reducing sedentariness and the other on increasing physical activity. Office workers (*n* = 263) were cluster randomized into one of two 6-month intervention groups, or a control group. Time spent in sedentary behavior, light-intensity, moderate and vigorous physical activity, and time in bed were assessed using accelerometers and diaries, both for 24 h in total, and for work and leisure time separately. Cardiorespiratory fitness was estimated using a sub-maximal cycle ergometer test. Intervention effects were analyzed using linear mixed models. No intervention effects were found, either for 24-h behaviors in total, or for work and leisure time behaviors separately. Cardiorespiratory fitness did not change significantly. Despite a thorough analysis of 24-h behaviors using CoDA, no intervention effects were found, neither for behaviors in total, nor for work and leisure time behaviors separately. Cardiorespiratory fitness did not change significantly. Although the design of the multi-component interventions was based on theoretical frameworks, and included cognitive behavioral therapy counselling, which has been proven effective in other populations, issues related to implementation of and compliance with some intervention components may have led to the observed lack of intervention effect.

## 1. Introduction

The beneficial effects of physical activity (PA) and negative effects of extensive sedentary behavior (SED) on physical and mental health outcomes at the population level are well established [1,2]. However, we need to understand better how less active individuals can be supported in changing their movement behaviors. In the context of movement behavior research, office workers are of particular interest because they are at high risk of accumulating large amounts of SED throughout the day, at work and during leisure [3,4,5]. Numerous studies have tried to identify effective strategies that help office workers to improve their movement behavior by either decreasing SED, increasing PA, or combining both of these, in order to improve various health outcomes [6,7,8,9]. However, studies show inconsistent results, partly because of poor study designs [9]. This needs to be rectified to support the development of evidence-based SED and PA recommendations for office workers [10].

First, holistic interventions designed to address movement behavior on multiple levels, i.e., both the individual, environmental and organizational levels, are considered to be more effective than interventions addressing only one of these levels [11,12]. Such multi-component interventions have been tried for the purpose of reducing SED among office workers, but with inconsistent results [9,13]. Few studies exist and the quality of this evidence has been rated as very low [9]. Concerning the individual level, cognitive behavioral therapy (CBT) techniques in combination with motivational interviewing (MI) have been used widely to support people in changing various health behaviors [14,15,16,17]. However, they have not yet been used as part of multi-component workplace interventions focusing on movement behavior among healthy office workers. CBT strategies focus on providing people with concrete tools for achieving and sustaining behavior change by supporting their intrinsic motivation and self-regulation skills [18]. Previous research has shown that people who are motivated by their own needs and desires find it easier to sustain new behaviors [19].

Several systematic reviews on workplace PA and SED interventions have pointed out that it is important to understand that changes in time spent in one behavior necessarily lead to changes in one or more other behaviors [9,20] because a day is constrained to 24 h However, previous studies have traditionally investigated effects of movement behavior interventions without taking the compositional nature of time-use data into account [9,20]. Even if an intervention targets only one behavior, this behavior has to be understood relative to all other behaviors, i.e., as part of a composition [21]. Treating time-use variables as absolute rather than relative data in regression analyses can potentially lead to misleading estimates of effect sizes (ES) [22].

Compositional data analysis (CoDA) is one statistical approach for handling time-use data [21,23,24]. Thus, CoDA allows analyses of intervention effects on the entire composition of different movement behaviors, such as SED, light-intensity physical activity (LIPA), moderate physical activity (MPA), vigorous physical activity (VPA) and time in bed [25]. Although there is significant potential for the use of CoDA for movement behavior research in occupational settings [21], very few studies have made use of it in office settings [26,27]. Furthermore, few workplace intervention studies have considered possible spill-over effects between work and leisure time behaviors [27] in their design and analyses [9]. It has been suggested that interventions targeting single behaviors while letting others change freely may lead to compensatory effects in order to maintain an overall stable level of PA or energy expenditure over time [28]. Thus, the total time spent in relevant behaviors for work and leisure combined might change marginally, even though notable changes would occur in each of these domains separately [9]. The actual effects of the intervention may therefore only be fully understood by analyzing both total and domain-specific behavior effects.

Investigating 24-h movement behavior requires that behavior can be measured in a reliable, valid and feasible way, such as with accelerometers [9]. Several systematic reviews have pointed to the need for more studies based on device-measured movement behavior rather than on self-reported data, which has previously been the dominant approach [6,7].

Addressing these research gaps, the present study used data from a three-armed 6-month cluster randomized controlled trial (RCT), examining the effects of two multi-component interventions that aimed at improving mental health and cognitive functioning among office workers by either decreasing SED or increasing PA during work and leisure time [29]. Improving cardiorespiratory fitness was a secondary aim. Intervention components were designed to influence behavior at the individual, environmental and organizational level. Regarding the individual level, this RCT was the first to address movement behavior among office workers using a combination of cognitive behavioral therapy and motivational interviewing. One intervention focused on reducing SED mainly by replacing it with LIPA and the other focused on increasing moderate to vigorous physical activity (MVPA). The interventions were delivered at the workplace but aimed at encouraging participants to change behavior throughout the entire day, during both work and leisure. An effectiveness analysis according to the published study protocol [29] has been published, showing no significant changes in primary outcomes, i.e., device-measured MVPA and SED [30]. This analysis did not apply CoDA, and domain-specific (work vs. leisure) effects were not investigated.

Thus, to provide a more in-depth examination of intervention effects, the first aim of this study was to determine intervention effects on the 24-h composition of movement behaviors, i.e., VPA, MPA, LIPA, SED and time in bed. The second aim was to investigate the extent to which domain-specific effects occurred. In these analyses, a CoDA approach was applied. The third aim was to determine the extent to which the interventions had effects on cardiorespiratory fitness.

## 2. Materials and Methods

This study was part of the research project “Physical activity and healthy brain functions”. Ethical approval was granted by The Stockholm Regional Ethical Review Board (2017/2409–31/1). All participants provided written informed consent before the first data collection. The study was conducted in accordance with the CONSORT guidelines for cluster RCTs (http://www.consort-statement.org/, accessed on 14 November 2017).

The trial was prospectively registered as ISRCTN92968402 on 27 February 2018, and recruitment started on 15 March 2018 (https://doi.org/10.1186/ISRCTN92968402, accessed on 13 April 2021). Data collection was performed between April 2018 and May 2019. The published study protocol contains a detailed description and rationale of the trial [29], of which the most important details are provided below. Participants did not receive any compensation for their participation in the study. However, they were allowed to take part in the intervention and measurements during working hours at the workplace.

### 2.1. Study Population

Office workers (*n* = 2033) from two Swedish companies were invited to participate in the RCT. Inclusion criteria were age 18–70 years and the ability to stand and to exercise. Persons who were very physically active, i.e., spent more than 30 min/day in MVPA in bouts of at least 10 min assessed using accelerometers, were excluded to focus on less active persons. Based on a priori power calculations, the aim was to recruit 330 participants [29]. However, only 298 persons volunteered and 263 of those were eligible for participation (see Figure 1).

Twenty-two clusters of eligible participants (*n* = 263) were randomized into one of the three arms, i.e., one intervention group focused on increasing MVPA (iPA), one intervention group focused on reducing SED (iSED) and a wait-list control group. Clusters were constructed based on participants having (1) the same team or line manager, (2) regular group meetings, and (3) limited regular meetings with other teams to limit spill-over effects. All participants already had access to height-adjustable tables when entering the study.

### 2.2. Interventions

The two interventions, iPA and iSED, included multiple components intended to influence behavior at different levels, i.e., the individual, the environmental and the organizational level, based on ecological models for health behavior [11,31] with the ultimate goal of improving mental health and cognition (see Figure 2). Participants decided together with coaches which type of activities would accommodate their needs and preferences in order to achieve a sustainable behavior change [18]. The CBT and MI techniques used in this study were: (1) goal setting tied to internal rewards and values and identification of the individual’s resources and boundaries for making behavior changes; (2) functional analysis including antecedents and consequences of undesired and desired behavior; (3) acceptance techniques for handling negative emotions; and (4) feedback on movement behaviors. Both interventions lasted for six months and were similar in design, while focusing on either increasing PA or reducing SED. A team leader was appointed to each cluster and instructed to implement a set of intervention components as described below (Figure 2). Three counselling sessions were individual and two were group sessions.

### 2.3. Control Group

The wait-list group was left unattended during the 6-month intervention period but received one of the two interventions after the follow-up measurement.

### 2.4. Data Collection and Processing

Demographic information, including age, sex and years of education, was assessed as part of an online questionnaire, which participants filled out at baseline and at the 6-month follow-up. The staff involved in data collection were blinded to participants’ group allocation.

### 2.5. Movement Behaviors

For this paper, we analyzed time spent in VPA, MPA, LIPA and SED, which was assessed using tri-axial ActiGraph™ GT3X accelerometers (ActiGraph LLC, Pensacola, FL, USA) at baseline and 6-month follow-up. Participants were instructed to wear the accelerometer on the hip for seven consecutive days during all waking hours and to move it to the left wrist when going to bed, removing it entirely only during water activities. To distinguish wake time from time in bed and work from leisure, participants were asked to fill out a diary throughout the measurement period. Participants were asked to note every day when they began trying to fall asleep and when they got out of bed. We used this measure of time in bed rather than time in sleep determined from the accelerometer recordings because the time it takes to fall asleep and occasional awakenings after sleep onset are normal, often healthy, parts of the sleep–wake cycle [32].

In case diary information was missing, a standard wake time from 6 AM to 11 PM was assumed. This standard wake time was applied to 1% of all days with valid data at baseline and to 3% of the days at follow-up.

Non-filtered, raw accelerometer data were processed to give a PA intensity metric based on a 10 Hz low-pass filter instead of the proprietary 1.6 Hz ActiGraph low-pass filter [33]. The 10 Hz method has shown higher validity in identifying high intensity PA [33,34]. In addition, 10 Hz filtered data were more strongly related to markers of cardio metabolic health than data filtered at 1.6 Hz [35]. An epoch length of three seconds was applied and outputs from the three separate accelerometer axes were combined to a vector magnitude. Non-wear time was defined as continuous zero output for at least 60 min with allowance of up to two minutes of output above zero but below the sedentary cut-point (1.5 metabolic equivalents (METs) [36]). A participant’s accelerometer data were considered valid if they contained at least four days with at least 10 h of wear time during waking hours. Seven participants were excluded due to having <4 valid days.

Wear time was then classified into different intensity levels in terms of energy expenditure, i.e., SED (<1.5 METs), LIPA (1.5–3 METs), MPA (3–6 METs) and VPA (>6 METs) [33]. We assumed that the distribution of time into different behaviors would be equal across wear time and non-wear time on the valid days with at least 10 h of wear time. Thus, time in the different behaviors was scaled to the participant’s total wake time. Accelerometer data were processed using MATLAB R2020a (MathWorks, Natick, MA, USA).

On average, participants had valid accelerometer data for 5.9 days (standard deviation (SD) 0.3) at baseline and 6.7 days (SD 0.8) at follow-up; they wore the device on average 96% (SD 4%) of their wake time at baseline and 95% (SD 6%) at follow-up with no difference across groups.

For the domain-specific analyses (aim 2), a participant’s accelerometer data were considered valid if he/she had at least two work days during the measurement period with at least 300 min of wear time at work.

### 2.6. Cardiorespiratory Fitness

Maximal oxygen uptake (VO2 max) was estimated using a sub-maximal cycle ergometer test [37], and was expressed both in relative terms (mL per minute per kg body mass) and as absolute values (L/min). Participants exercised on a cycle ergometer (model 828E, Monark, Varberg, Sweden) for four minutes on a low intensity, standardized to 0.5 kilopounds at a pedaling rate of 60 rounds per minute. Thereafter, an individualized higher work intensity was set, and the participant biked for an additional four minutes. The difference in heart rate divided by the difference in work intensity was used in the formula presented by Björkman et al. 2016 [37].

### 2.7. Compositional Data Analysis

Because times spent in VPA, MPA, LIPA, SED, and in bed during a day add up to a constant value (i.e., 1440 min), they are compositional. Thus, we applied CoDA, as introduced above [21,24].

To address aim 1, each participant’s daily time use was conceptualized as a composition consisting of average daily time spent in VPA, MPA, LIPA, SED, and time in bed. Compositional (geometric) means of time spent in these five behaviors were calculated across days and linearly scaled so that total daily time spent in all behaviors added up to an entire day, i.e., 1440 min. The composition of each participant was then transformed into a set of four isometric log-ratio coordinates (ilrs) [21,38], as exemplified below for VPA.



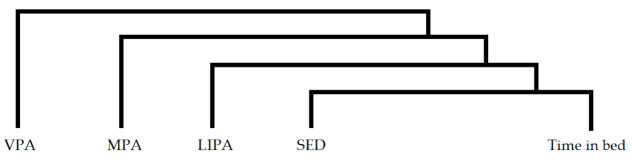

(1)$$\begin{array}{l} ilr_{1}=\sqrt{\frac{4}{5}}\ln\left(\frac{VPA}{\sqrt[{4}]{MPA*LIPA*SED*Time\;in\;bed}}\right)\\ ilr_{2}=\sqrt{\frac{3}{4}}\ln\left(\frac{MPA}{\sqrt[{3}]{LIPA*SED*Time\;in\;bed}}\right)\\ ilr_{3}=\sqrt{\frac{2}{3}}\ln\left(\frac{LIPA}{\sqrt[{2}]{SED*Time\;in\;bed}}\right)\\ ilr_{4}=\sqrt{\frac{1}{2}}\ln\left(\frac{SED}{Time\;in\;bed}\right) \end{array}$$


In this case, ilr_1_ expresses the ratio of time spent in VPA to time spent in all other behaviors. A separate set of four ilr coordinates was created for each of the five movement behaviors by rotating the position of behaviors in the equation, so that each of the five behaviors was expressed relative to the geometric mean of the remaining behaviors [39].

To answer aim 2, a participant’s daily time use was expressed as a composition consisting of nine behaviors: VPA, MPA, LIPA and SED during work time, and VPA, MPA, LIPA, SED and time in bed during leisure. Nine sets of eight ilrs were then determined for each of the nine behaviors, following the same rotating principle as described above, expressing each behavior in relation to all others (see Appendix A).

For both aim 1 and aim 2, the intervention effect on time spent in one behavior relative to time in all other behaviors was assessed by using the first ilr (ilr_1_) as the outcome variable in linear mixed models (see below).

### 2.8. Statistical Analyses

At baseline, 263 persons were included in the RCT. We performed a complete case analysis because the previously published effectiveness analysis found no differences between results based on complete case and intention to treat analyses [30]. Thus, three different analytic samples were created for the three different analyses addressing aims 1–3. Participants with complete data for movement behaviors at baseline and follow-up, i.e., VPA, MPA, LIPA, SED and time in bed, were included in the analysis of 24-h movement behavior (*n* = 158) (aim 1). Two participants did not participate in baseline measurement, 96 did not participate at follow-up, and 7 participants had <4 valid days.

Participants with complete data for movement behaviors during both work, i.e., VPA, MPA, LIPA, SED, and leisure, i.e., VPA, MPA, LIPA, SED and time in bed, were included for the domain-specific analysis (*n* = 150) (aim 2). Seven participants were excluded due to having less than two measurement days with sufficient work time. One participant was excluded because of change in employment status.

Participants with complete data for cardiorespiratory fitness were included in the analysis to address aim 3 (*n* = 151).

A dropout analysis was performed, comparing age, years of education, body mass index (BMI, kg/m^2^), cardiorespiratory fitness and movement behaviors at baseline between participants dropping out and those completing the intervention, using Student’s t-tests; differences in sex distribution were checked using a chi-square test.

Standard summary statistics (i.e., mean, SD and proportions) were calculated to describe key demographic characteristics of the three analytic samples, i.e., age, sex and years of education.

Then, we investigated and compared age, years of education and BMI between the iPA, iSED and control groups at baseline using Student’s t-tests; differences in sex distribution between groups were checked using a chi-square test.

Next, one linear mixed effects model was fitted for all outcomes, i.e., each of the movement behaviors (aims 1 and 2) and for cardiorespiratory fitness (aim 3), to examine possible intervention effects. In these models, ilr_1_ for each movement behavior, and cardiorespiratory fitness were used as outcome variables. These models accounted for the random effect of individual within each cluster and estimated the fixed effects of intervention group (3 levels), time (2 levels), and interaction between intervention group and time (3 × 2 levels). We applied an unstructured covariance structure to the models. Baseline age, sex and education were included as covariates. Group-specific effects for all outcomes were analyzed by comparing the intervention groups to each other and to the control group, using pairwise comparisons (Tukey). Baseline differences between groups and within-group changes for the outcome variables were investigated in the same manner. Normal distribution of residuals was visually inspected post hoc and confirmed that the normality assumptions for linear regression were fulfilled. A *p*-value of ≤0.05 was set as the level of statistical significance.

An intention to treat analysis including all participants (*N* = 263) was performed as a sensitivity analysis for all outcomes.

We performed an additional analysis to compare both intervention groups combined to the control group to explore potential general intervention effects on the various outcomes.

All analyses were performed in R [40], CoDA analyses were performed using the Compositions package [41] and linear mixed models were performed using the lme4 package [42]. The researchers performing the analyses were blinded to the participants’ group affiliation until the analyses were completed.

## 3. Results

### 3.1. Demographic Characteristics

Figure 1 presents a flow diagram for enrollment, participation and analysis.

Baseline characteristics on age, sex, years of education and BMI of participants included in the three different analytic samples are reported in Table 1. Participants included in the analysis of movement behavior were on average 43 (SD 8) years old, 23% were male, they had on average 15 (SD 2) years of education and had a BMI of 25.2 (SD 4.2). The other two samples were very similar to this.

Within each of the three analytic samples, dropouts were 1 year younger, had 1 year shorter duration of education and were more likely to be part of the iSED group compared to participants that did not drop out. In the analytic sample for movement behaviors (aim 1), time spent in SED was larger among dropouts. Detailed results from the dropout analysis are presented in Appendix D, Table A7.

### 3.2. Intervention Effects on 24-h Movement Behavior

Table 2 and Table 3 show compositional means for minutes spent in each behavior at baseline and follow-up for the intervention groups and controls, for the overall 24-h behavior (Table 2), and for domain-specific behaviors (Table 3). Very small changes in behaviors occurred between baseline and follow-up in all groups, irrespective of domain.

At baseline, groups did not differ in movement behaviors, as expressed by ilr_1_ for each behavior, i.e., the time spent in that specific behavior relative to time in all other behaviors. This was observed both for overall 24-h behaviors (aim 1), and domain-specific behaviors (work vs. leisure, aim 2).

Figure 3 shows marginal plots, based on the linear mixed models, analyzing effects on 24-h movement behavior (aim 1). No significant intervention effects (i.e., interaction between time and group) were found for any of these movement behaviors. The intervention effect estimates were very small, consistent with the changes in compositional means in Table 2. Detailed information on estimates and confidence intervals (CI) can be found in Appendix B, Table A1.

Figure 4 shows marginal plots, based on the linear mixed models, analyzing effects on domain-specific movement behavior (aim 2). Corroborating the analysis for aim 1, no significant intervention effects (i.e., interaction between time and group) were found for any movement behavior. The intervention effect estimates were very small, consistent with the compositional means in Table 3. Detailed information on estimates and confidence intervals (CI) can be found in Appendix B, Table A2. We did not find any significant within-group changes.

The sensitivity analysis based on all included participants (*N* = 263) confirmed the lack of intervention effect on movement behaviors. The additional analyses comparing both intervention groups combined with the control group showed no general intervention effect on any behavior, not even when domain-specific effects were considered. Detailed results can be found in Appendix C, Table A3, Table A4, Table A5 and Table A6.

### 3.3. Intervention Effects on Cardiorespiratory Fitness

Table 4 shows baseline and follow-up values for cardiorespiratory fitness. At baseline, cardiorespiratory fitness was higher for participants in iSED compared to iPA (ES 3.17, 95% CI: 0.25 to 6.08, *p* = 0.028), but only when expressed as mL·kg^−1^·min^−1^.

No intervention effects were found for cardiorespiratory fitness (Figure 5). Estimated estimates with 95% CI can be found in Table 5.

A significant increase in cardiorespiratory fitness occurred only within the iPA group (mL·kg^−1^·min^−1^: ES 1.44, 95% CI: 0.57 to 2.32, *p* < 0.001; L/min: ES 0.12, 95% CI: 0.05 to 0.18) from baseline to 6-month follow up. The sensitivity analysis based on all included participants (*N* = 263) confirmed the lack of intervention effect on fitness. The additional analyses of participants in both intervention groups combined versus the control group showed that no general intervention effect on cardiorespiratory fitness had occurred (mL·kg^−1^·min^−1^: ES 0.73, 95% CI: −0.20 to 1.68; L/min: ES 0.035 95% CI: −0.02 to 0.12)).

## 4. Discussion

The aim of this study was to investigate the effectiveness of two multi-component interventions among office workers in changing 24-h movement behavior, i.e., the composition of VPA, MPA, LIPA, SED and time in bed, applying a CoDA approach (aim 1). We also examined possible domain-specific effects (aim 2) on movement behaviors at work and in leisure. In addition, we assessed intervention effects on cardiorespiratory fitness (aim 3). Neither of the two interventions were successful in changing movement behaviors or cardiorespiratory fitness, compared to the control group.

### 4.1. Comparisons with Previous Studies

Our results confirm the previous effectiveness analysis performed according to the published study protocol [29], which did not find any intervention effect on average daily %MVPA or %SED [30]. The in-depth analysis in the present study, examining domain-specific effects and using CoDA, also did not disclose any relevant intervention effect. In addition, we were able to overturn some of our preliminary suggestions for an explanation of the negative results. First, no compensation effect appeared to occur across domains, i.e., work and leisure, which could otherwise have led to a net to null effect; the interventions appeared to be equally ineffective in changing behaviors in both domains. Furthermore, we distinguished between MPA and VPA in this study and concluded that the null effect on MVPA found in the previous effectiveness study [30] was not due to the fact that participants increased one of these sub-behaviors while decreasing the other.

To our knowledge, this workplace RCT intervention study aiming at changing movement behavior is among the first to investigate intervention effects using CoDA, which is a comprehensive and statistically sound approach for analyzing time-use data [21,23,24]. We have identified only one other study using CoDA for evaluating effects of a workplace RCT intervention targeting movement behavior [27]. That study aimed at reducing workplace sitting time by initiatives directed towards the individual (e.g., health coaching and motivational interviewing by trained health coaches); the workplace environment (e.g., sit–stand workstations); and the organization (e.g., management consultation and emails from worksite managers). Sitting time was significantly reduced after three months, mainly during work time where it was replaced by standing time, with no compensation effects occurring during non-work hours. In contrast to our study, participants in that study received sit–stand workstations at the beginning of the trial whereas participants in our study already had had sit–stand workstations for a long period, which is highly common in Sweden.

### 4.2. The Lack of Intervention Effects

Several aspects relating to the different intervention components might explain the findings. Multi-component interventions have previously been shown to be more effective in changing behavior compared to single-component interventions [9]. The interventions in our study were based on ecological models of health behavior and addressed behavior on multiple levels. Although the CBT and MI techniques used in this study have been proven effective in previous studies for changing [14,15,16,17] and sustaining [18] health behaviors, more research is needed to identify the potential of using CBT and MI as part of multi-component interventions aiming at improving mental health and cognitive functions by changing movement behavior, in particular among office workers.

Considering that the interventions were designed based on theoretical frameworks and previous evidence on the effectiveness of CBT and MI, implementation and measurement bias might explain the findings of this study. In the analysis of this study, we did not consider the participants’ compliance with the intended protocol, i.e., how many of the five counselling sessions a participant actually attended. Poor attendance might have led to a less effective intervention. However, per protocol analyses were performed in the previous study and confirmed a lack of intervention effect even among participants that had attended at least three of the five counselling sessions. Team leaders had a prominent role in the interventions [30]. They were responsible for implementing the environmental (standing or walking meetings, lunch walks and exercise sessions) and organizational (encouraging behavior change throughout the intervention period) components. However, we have no data on how successful team leaders were in practicing their tasks. For practical reasons, not all team leaders were senior management. A higher rank position among colleagues could likely have facilitated the leader role of more junior team leaders, and thus their impact on participants’ behaviors. In addition, the larger of the two participating companies moved to a new activity-based open space office building during the study period. Changing from fixed desks in close proximity to colleagues within the same cluster to this open space might have had a negative impact on the effectiveness of the environmental intervention components. Concerning the organizational component, a potential selection bias has to be considered [43]. Companies and employees represented a convenience sample, and the two participating companies were already very active in health promotion efforts.

Regarding potential measurement bias, device-measured PA and SED have been shown to be more valid than self-reported behaviors [44]. However, in this study we could not distinguish between different types of PA, such as cycling, weight-bearing exercise and swimming, because the accelerometers were hip-worn and their outcome was analyzed based on counts. Self-reported PA and inclinometer-measured PA might better capture detailed and relevant changes in such activities. It is possible that participants may, thus, have changed the contents of PA during the intervention period. However, previous analyses found that inclinometer-measured PA had not changed after the intervention [30]. In addition, our analysis of cardiorespiratory fitness did not show any significant intervention effect, suggesting that people did not change the contents of their PA in any way that would influence fitness.

Further possible explanations for the lack of an intervention effect might relate to participants’ movement behaviors when entering the study. Office workers are considered to be at high risk of accumulating large amounts of SED during both work and leisure, which identifies them as a relevant target for interventions addressing SED and PA. When including persons in this RCT, we faced a trade-off between achieving sufficient statistical power and enrolling only less-active persons who would likely benefit most from the intended interventions. Thus, persons who already spent more than 30 min/day in MVPA in bouts of at least 10 min were excluded. High levels of SED were not chosen as an inclusion criterion because almost all employees showed high levels of SED [45,46]. Participants in the study had quite high levels of SED (on average 399 min of daily work time at baseline, consistent with other studies [47], and 421 min during leisure). However, they also spent on average 34 min of daily work time and 77 min of leisure time in VPA and MPA, which is high compared to populations in other studies [48]. Thus, the baseline movement behavior of participants should be considered when interpreting the findings of this study.

Another factor may be that persons did change behavior sometime during the 6-month intervention period but did not sustain that change throughout. We could not examine this hypothesis further because fewer than 25% of participants joined the three-month accelerometry measurement, which was offered as an additional but not mandatory measurement to participants.

Dropout rates varied from around 27% in the iPA group and 38% in the control group to 57% in the iSED group (see Figure 1). The dropout analysis showed that persons that dropped out were significantly younger, and they had shorter duration of education compared to participants who did not drop out. However, the differences were small in absolute terms (43 vs. 41 years of age, 15 vs. 14 years of education). In addition, the proportion of men was significantly higher among those who dropped out compared to those that completed the intervention (32% vs. 23%). This might indicate that the interventions were less attractive to men, especially the iSED intervention. Further analyses are planned to investigate reasons for dropping out.

### 4.3. Strengths and Limitations

This study is among the first to apply a CoDA approach within occupational health research. We would like to emphasize the need for conceptualizing and understanding time spent in movement behaviors as a compositional structure. CoDA offers a useful and statistically sound approach for in-depth analysis of potential intervention effects on 24-h movement behavior [21,25].

Another strength of this study is that it was conducted in a real-life setting. Conclusions drawn from this study may thus be relevant for similar, future studies in office settings.

Another strength of our study was the use of objectively measured movement behaviors. The quality of our accelerometer data was high because of good wearing compliance. Consensus on optimal data collection and processing criteria has not been reached to date [49], but we used a novel method for filtering raw accelerometer data that overcomes some critical issues in data processing [33]. This included data processing based on a 10 Hz low-pass filter instead of the proprietary 1.6 Hz ActiGraph low-pass filter [33], and an epoch length of three seconds. This procedure enabled us to analyze higher-intensity activity data with better validity [33,35]. However, data analyzed with this method are not directly comparable to data analyzed with other methods. Using 3-s epochs when analyzing accelerometer data leads to an increase in minutes classified as higher intensity compared to using 60-s epochs.

Several limitations have to be recognized when interpreting the results. As mentioned above, data concerning implementation of and compliance with the intended intervention protocols were not considered. In addition, attention should be paid to the fact that participants included in this study were, on average, very well educated, female and 43 years old. Thus, results might not be representative for populations with other characteristics.

We used validated cut-points for classifying participants’ accelerometer-measured movement behavior into different physical intensity groups, reflecting energy expenditure. Although this is a feasible and commonly used approach, it may introduce some uncertainty in intensity classification because behaviors within a particular intensity group, i.e., SED, LIPA, MPA, VPA, may differ in intensity between participants due to differences in cardiorespiratory fitness level, body weight and other personal characteristics. This, in turn, may lead to some loss in statistical power.

## 5. Conclusions

The two investigated multi-component cluster RCT interventions were not successful in changing 24-h movement behavior in a sample of relatively active office workers, either during work or during leisure time. Nor did they result in increased cardiorespiratory fitness.

Although the design of the multi-component interventions was based on theoretical frameworks, and included cognitive behavioral therapy counselling, which has been proven effective in previous studies with other populations, issues related to implementation of and compliance with some intervention components may have led to the observed negative results. Future studies should consider these aspects and invest particular effort in enrolling less-active persons, because they are likely to benefit most from such interventions. Intervention studies should be designed and analyzed with due consideration to all movement behaviors occurring throughout the day. The CoDA approach provides a useful strategy for taking the compositional nature of 24-h movement behaviors into account when evaluating the effectiveness of interventions.

## Figures and Tables

**Figure 1 ijerph-18-04191-f001:**
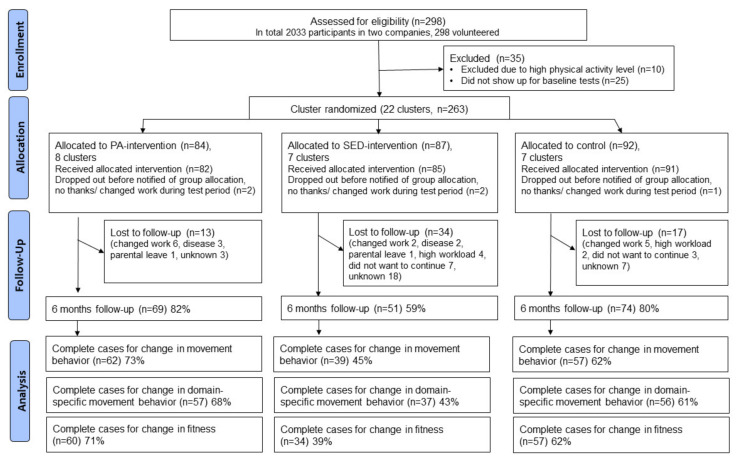
Flow diagram for enrollment, participation and analysis.

**Figure 2 ijerph-18-04191-f002:**
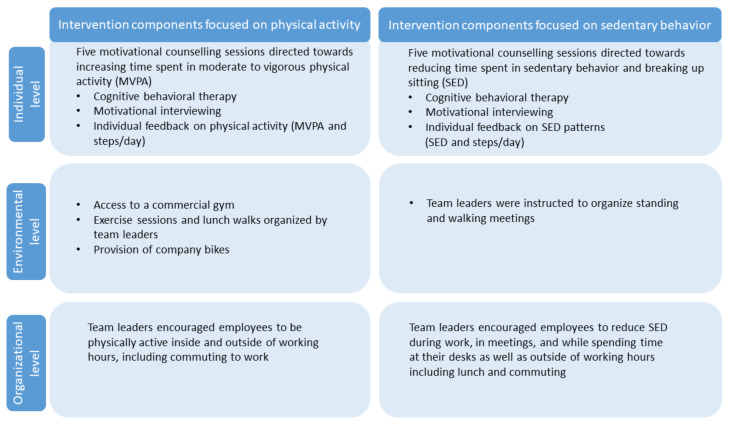
Overview of the intervention design showing which components were addressing the individual, environmental and organizational levels.

**Figure 3 ijerph-18-04191-f003:**
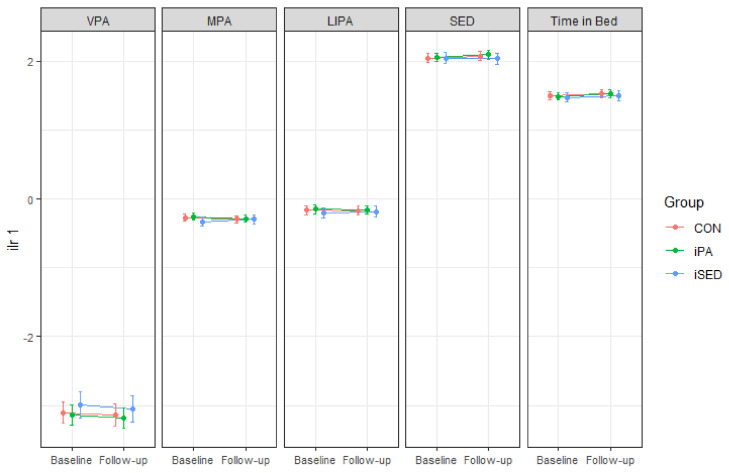
Marginal means (with 95% CI) for each movement behavior, expressed in terms of ilr_1_ (time spent in that behavior relative to time spent in all other behaviors) at baseline and 6-month follow-up. Values were estimated based on the results in the linear mixed models. VPA: vigorous-intensity physical activity, MPA: moderate-intensity physical activity, LIPA: light-intensity physical activity, SED: sedentary behavior. iPA: Intervention group focused on increasing physical activity, iSED: Intervention group focused on reducing sedentary behavior, CON: Control group.

**Figure 4 ijerph-18-04191-f004:**
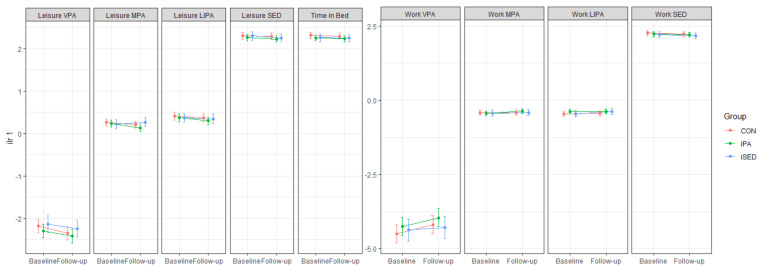
Marginal means (with 95% CI) for each domain-specific movement behavior, expressed in terms of ilr_1_ (time spent in that behavior relative to time spent in all other behaviors) at baseline and 6-month follow-up with 95% CI bars. Values were estimated based on the results in the linear mixed models. VPA: vigorous-intensity physical activity, MPA: moderate-intensity physical activity, LIPA: light-intensity physical activity, SED: sedentary behavior. iPA: Intervention group focused on increasing physical activity, iSED: Intervention group focused on reducing sedentary behavior, CON: Control group.

**Figure 5 ijerph-18-04191-f005:**
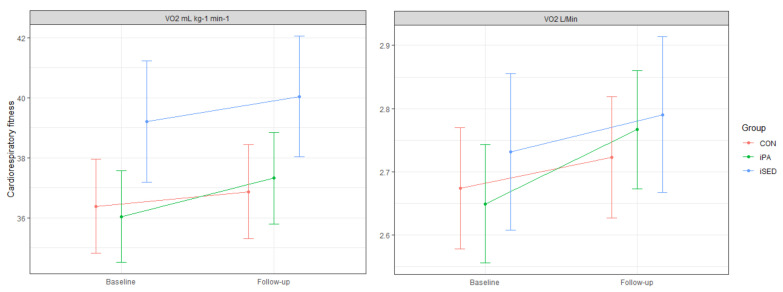
Marginal means (with 95% CI) for cardiorespiratory fitness, expressed as mL·kg^−1^·min^−1^ and as L/min, at baseline and 6-month follow-up. Values were estimated based on the results in the linear mixed models.

**Table 1 ijerph-18-04191-t001:** Participants’ baseline characteristics for the three analytic samples.

Complete cases for movement behavior analysis (aim 1)	**Demographic Characteristics**	**All Participants** ***n*** **= 158**	**iPA** ***n* = 62**	**iSED** ***n* = 39**	**Control Group** ***n* = 57**
	Age, years (mean (SD))	43 (8)	41 (9)	42 (8)	45 (7) ^a^
	Sex, men (*n* (%))	36 (23)	13 (21)	10 (26)	13 (23)
	Education, years (mean (SD))	15 (2)	15 (2)	15 (2)	15 (2)
	BMI, kg/m^2^ (mean (SD))	25.2 (4.2)	24.9 (4.0)	24.7 (3.7)	26.2 (4.5) ^b^
Complete cases for domain-specific movement behavior analysis (aim 2)		**All Participants** ***n* = 150**	**iPA** ***n* = 57**	**iSED** ***n* = 37**	**Control Group** ***n* = 56**
	Age, years (mean (SD))	43 (8)	41 (9)	41 (8)	45 (7) ^a^
	Sex, men (*n* (%))	37 (25)	13 (23)	10 (27)	10 (25)
	Education, years (mean (SD))	15 (2)	15 (2)	15 (2)	15 (2)
	BMI, kg/m^2^ (mean (SD))	25.2 (4.2)	24.9 (4.1)	24.0 (2.7)	26.2 (4.5) ^b^
Complete cases for cardiorespiratory fitness analysis (aim 3)		**All Participants** ***n* = 151**	**iPA** ***n* = 60**	**iSED** ***n* = 34**	**Control Group** ***n* = 57**
	Age, years (mean (SD))	43 (8)	42 (9)	42 (8)	45 (8)
	Sex, men (*n* (%))	39 (26)	13 (22)	11 (32)	15 (26)
	Education, years (mean (SD))	15 (2)	15 (2)	15 (2)	15 (2)
	BMI, kg/m^2^ (mean (SD))	25.2 (4.1)	24.7 (3.6)	24.2 (3.9)	26.3 (4.5) ^a,b^

iPA: Intervention group focused on increasing physical activity, iSED: Intervention group focused on reducing sedentary behavior; ^a^ Significantly different from participants in iPA (*p* < 0.05); ^b^ Significantly different form participants in iSED (*p* < 0.05).

**Table 2 ijerph-18-04191-t002:** Baseline and 6-month follow-up values for minutes spent in 24-h movement behaviors (compositional mean and % of 24-h in parentheses).

	All Participants *n* = 158		iPA *n* = 62		iSED *n* = 39		Control Group *n* = 57	
	Baseline	Follow-Up	Baseline	Follow-Up	Baseline	Follow-Up	Baseline	Follow-Up
VPA	8 (0.5)	7 (0.5)	8 (0.5)	7 (0.5)	9 (0.6)	7 (0.5)	8 (0.6)	7 (0.5)
MPA	95 (6.6)	93 (6.4)	97 (6.7)	92 (6.4)	92 (6.4)	96 (6.7)	95 (6.6)	93 (6.5)
LIPA	106 (7.3)	103 (7.2)	107 (7.4)	103 (7.2)	103 (7.1)	106 (7.4)	105 (7.3)	103 (7.2)
SED	767 (53.3)	768 (53.4)	766 (53.2)	771 (53.6)	773 (53.7)	764 (53)	769 (53.4)	768 (53.3)
Time in bed	464 (32.3)	469 (32.5)	463 (32.1)	467 (32.4)	464 (32.2)	467 (32.4)	463 (32.2)	468 (32.5)

**Table 3 ijerph-18-04191-t003:** Baseline and 6-month follow-up values for minutes spent in domain-specific 24-h movement behaviors (compositional mean and % of 24-h in parentheses).

	All Participants *n* = 150		iPA *n* = 57		iSED *n* = 37		Control Group *n* = 56	
	Baseline	Follow-Up	Baseline	Follow-Up	Baseline	Follow-Up	Baseline	Follow-Up
Leisure VPA	6 (0.4)	5 (0.4)	6 (0.4)	5 (0.4)	7 (0.5)	6 (0.4)	6 (0.4)	5 (0.4)
Leisure MPA	61 (4.2)	60 (4.2)	63 (4.3)	58 (4.0)	61 (4.2)	65 (4.5)	60 (4.2)	59 (4.1)
Leisure LIPA	70 (4.8)	68 (4.8)	71 (4.9)	68 (4.7)	69 (4.8)	69 (4.8)	70 (4.9)	69 (4.8)
Leisure SED	421 (29.2)	418 (29.0)	419 (29.1)	415 (28.8)	430 (29.8)	420 (29.2)	417 (29.0)	421 (29.2)
Work VPA	1 (0.05)	1 (0.07)	1 (0.06)	1 (0.09)	1 (0.06)	1 (0.04)	1 (0.04)	1 (0.04)
Work MPA	33 (2.3)	34 (2.4)	33 (2.3)	36 (2.5)	33 (2.3)	34 (2.4)	32 (2.2)	33 (2.3)
Work LIPA	33 (2.3)	34 (2.4)	31 (2.1)	33 (2.3)	34 (2.4)	35 (2.4)	33 (2.3)	35 (2.4)
Work SED	399 (27.7)	399 (27.7)	400 (27.8)	397 (27.6)	401 (27.8)	406 (28.2)	398 (27.7)	399 (28.9)
Time in bed	417 (28.9)	420 (29.2)	422 (29.3)	423 (29.4)	414 (28.7)	417 (28.9)	414 (28.7)	418 (29.0)

**Table 4 ijerph-18-04191-t004:** Baseline and 6-month follow-up values for cardiorespiratory fitness (mean (SD)).

Cardiorespiratory Fitness	All Participants *n* = 151		iPA *n* = 60		iSED *n* = 34		Control Group *n* = 57	
	Baseline	Follow-Up	Baseline	Follow-Up	Baseline	Follow-Up	Baseline	Follow-Up
mL·kg^−1^·min^−1^	36.9 (7.4)	37.8 (7.4)	36.6 (7.0)	37.8 (7.2)	39.8 (7.5)	40.7 (7.2)	35.5 (7.5)	35.9 (7.4)
L/min	2.7 (0.60)	2.8 (0.6)	2.6 (0.6)	2.8 (0.6)	2.8 (0.6)	2.9 (0.6)	2.6 (0.6)	2.7 (0.6)

**Table 5 ijerph-18-04191-t005:** Differences in change from baseline to follow-up between groups for cardiorespiratory fitness (estimates with 95% CI), based on the linear mixed models.

Cardiorespiratory Fitness	Comparing iPA to the Control Group	Comparing iSED to the Control Group	Comparing iPA to iSED
**mL·kg^−1^·min^−1^**	0.96 (−0.26 to 2.19)	0.34 (−1.10 to 1.78)	0.61 (−0.81 to 2.04)
**L/min**	0.07 (−0.02 to 0.16)	0.01 (−0.09 to 0.11)	0.05 (−0.04 to 0.16)

## Data Availability

The data related to this study are available upon reasonable request.

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
