# Peer review of "Effects of Two Randomized and Controlled Multi-Component Interventions Focusing On 24-Hour Movement Behavior among Office Workers: A Compositional Data Analysis"

_ijerph, 2021, doi:10.3390/ijerph18084191_

Round 1

Reviewer 1 Report

A large part of the reference list is missing [22]/[41]. That hampers reviewing.

The authors apply mixed modelling techniques. The advantage of mixed is that it can handle incomplete cases. However, they only analyse complete cases, referring to a study [27], that cannot be checked because the reference is missing. Generally it is strongly advised to analyse all cases, see e.g. Little & Rubin (1987).

It seems that there also was a mid-term measure (line 461), but too few measures were taken. How many is too few? As incomplete cases can be included in mixed modelling, it can be considered too include this mid-term in the analyses.

There is a large dropout , up to 60%. The authors should perform a dropout analysis; compare baseline measures of retained participants and dropouts. Note that it is not sufficient to conclude that there are no differences when p-values are >0.05. Effect sizes should also be examined.

The dropout is significantly larger in the iSED group. Do the authors have an explanation for that?

It is not clear how the mixed models are formulated. How many levels are postulated? Which covariance structure?

The authors do not find any significant differences in effect between the groups. Did the authors perform an a-priory power analysis? Why did they include 298 participants?

Please provide an abbreviation list.

Little, R. J. A., & Rubin, D. B. (1987). Statistical analysis with missing data. New York: John Wiley and Sons.

Author Response

Reviewer 1:

Thank you for your valuable comments.

A large part of the reference list is missing [22]/[41]. That hampers reviewing.

Answer: We apologize for this mistake. We realized that a bug had hit our reference list and we have now updated the list of references.

The authors apply mixed modelling techniques. The advantage of mixed is that it can handle incomplete cases. However, they only analyse complete cases, referring to a study [27], that cannot be checked because the reference is missing. Generally it is strongly advised to analyse all cases, see e.g. Little & Rubin (1987).

Answer: Several arguments have led us to choose complete case analysis: as you mention, we refer to study 27 (Nooijen et al.). This is the previously published study by our group in which effectiveness of the intervention was investigated. Even though slightly different outcomes were considered in this previous study, no difference between intention to treat analysis and complete cases analysis was found. This motivated us to choose the more robust method of using complete cases only.

Indeed, one advantage of mixed linear models is that it can handle missing values. However, this is not the only advantage. It also allowed us to take cluster and subjects as random effects into account in the statistical models.

It seems that there also was a mid-term measure (line 461), but too few measures were taken. How many is too few? As incomplete cases can be included in mixed modelling, it can be considered too include this mid-term in the analyses.

Answer: Thanks for this relevant question. We would have liked to use the 3-month accelerometry measurement for testing our hypothesis that a behavior change has occurred during the intervention but was not sustained until the end. However, as we stated, too few participants joined this measurement. This measurement was offered to participants, but it was not as “mandatory” as the baseline and follow-up measurements. Since participation rate was so low, this data was not considered for analysis.

We have made slight changes to the manuscript to provide the readers with some more information:

“We could not examine this hypothesis further since less than 25% participants joined the three-months accelerometry measurement which was offered as an additional, but not mandatory measurement to participants in the intervention groups.” (Lines 485-488)

There is a large dropout, up to 60%. The authors should perform a dropout analysis; compare baseline measures of retained participants and dropouts. Note that it is not sufficient to conclude that there are no differences when p-values are >0.05. Effect sizes should also be examined.

Answer: We have now added a description of the dropout analysis to the statistical analysis section as well as the result section.

“A dropout analysis was performed, comparing age, years of education, body mass index (BMI), cardiorespiratory fitness and movement behaviors at baseline between participants dropping out and those completing the intervention, using Student’s t-tests; differences in sex distribution were checked using Chi-square test.” (Lines 272-275)

 “Within each of three analytic samples, dropouts were 1 year younger, had 1 year shorter duration of education and were more likely to be part of the iSED group compared to participants that did not drop out. In the analytic sample for movement behaviors (aim 1) time spent in SED was larger among dropouts. Detailed results from the dropout analysis are presented in Appendix D, Table 7.” (Lines 316-320)

Table 7 in Appendix D displays detailed results of the dropout analysis, including effect sizes and confidence intervals.

The dropout is significantly larger in the iSED group. Do the authors have an explanation for that?

Answer: We have now added the following sentences to the discussion section regarding the higher dropout within the iSED groups:

“Dropout rates varied from around 27% in the iPA group and 38% in the control group to around 57% in the iSED group (see Figure 1). The dropout analysis showed that persons that dropped out were significantly younger, and they had shorter duration of education compared to participants who did not drop out. However, the differences were small in absolute terms (43 vs. 41 years of age, 15 vs. 14 years of education). Also, the proportion of men was significantly higher among those who dropped out compared to those that completed the intervention (32% vs. 23%). This might indicate that the interventions were less attractive to men, especially the iSED intervention. Further analyses are planned to investigate reasons for dropping out.” (Lines 489-497)

We have also performed a qualitative study investigating acceptability and feasibility among participants, coaches and people responsible for setting up the intervention in the companies. Preliminary and not yet published results of this study indicate that participants perceived the iPA group as more attractive because 1) iPA participants received free access to a gym and 2) some participants expressed that they felt that they wanted to go all in with changing their behavior and actually wanted to get help with starting to exercise and “not only” change their sedentary behavior. This might be because it is socially more desirable to exercise and be active than “just” making sure not to sit too much. This might be especially true for younger men.  So even though reducing sedentary time might be associated with less investment of resources for achieving a successful behavior change (in terms of energy and time for example), increasing physical activity might be more desirable.

It is not clear how the mixed models are formulated. How many levels are postulated? Which covariance structure?

 Answer: We have added the number of levels and information on type of covariance structure in the manuscript as follows:

“These models accounted for the random effect of individual within each cluster and estimated the fixed effects of intervention group (3 levels), time (2 levels), and interaction between intervention group and time (3 x 2 levels). We applied an unstructured covariance structure to the models.”  (Lines 285-288)

The authors do not find any significant differences in effect between the groups. Did the authors perform an a-priory power analysis? Why did they include 298 participants?

Answer: The target was to include 330 persons, based on sample size calculations which have been described in detail in the published study protocol (Nooijen et al. 2019). We have now added two sentences to mention the a-priori sample size calculations with reference to the published study protocol

 “Based on a priori power calculations, the aim was to recruit 330 participants [26]. However only 298 employees volunteered and 263 of those were eligible for participation (see Figure 1).” (Lines 141-143)

In addition, the Flow diagram for enrollment, participation, and analysis in figure 1 shows that 298 persons volunteered for the study and why 35 persons had to be excluded. We have moved the flow chart for enrollment (Figure 1) to the section study population so that the reader can early on understand how and why employees were enrolled into the study. The screenshot below shows the upper part of the flow diagram answering your question:

Please provide an abbreviation list.

 Answer: According to the IJERPH instructions for Authors, no abbreviation list is required. As instructed, we have defined abbreviations in parentheses the first time they appear in the text. If the editor would anyway like us to add an abbreviation list, we can of course provide one. 

Little, R. J. A., & Rubin, D. B. (1987). Statistical analysis with missing data. New York: John Wiley and Sons.

Reviewer 2 Report

This is an interesting study about Two Randomized and Controlled Multi-Component 2 Interventions. The novelty of this study is based on a new statistical method. Methods should be used to solve problems, not creating new ones. However, the approach is interesting. Upon that, there are some questions this reviewer would like to see clarified.

L.56: The authors may indicate the issues of this poor study designs. Why they are poor? Moreover, the authors state "studies" but only one reference was included.

L.98-99: for how long the interventions. The introduction did not highlight a research gap related to interventions duration.

L131. Not knowing the mean age is hard to support the findings. How can age affect that? Also, there are no information regarding sample characteristics, such as BMI. This will difficult the results discussion.

Table 1. This reviewer believes that this information must be in methods regarding the sample characteristics.

This reviewer did not clearly understood the intervention. Was this multicomponent "training"? Exercise? Physical activity? What type of exercises?

Discussion: Based on the interventions characteristics, the results are prone to be predicted. PA sessions are not training or exercise, some aspects such as load and individualization are not controlled. How can that explain the results? Please highlight.

L404: should be ) instead of =

Author Response

Reviewer 2:

This is an interesting study about Two Randomized and Controlled Multi-Component 2 Interventions. The novelty of this study is based on a new statistical method. Methods should be used to solve problems, not creating new ones. However, the approach is interesting. Upon that, there are some questions this reviewer would like to see clarified.

 Answer: Thank you for your valuable comments.

L.56: The authors may indicate the issues of this poor study designs. Why they are poor? Moreover, the authors state "studies" but only one reference was included.

Answer: The reference is the Cochrane Review by Shrestha et al. 2018, which summarizes a large range of studies. Therefore we used “studies” in plural.

In the paragraphs following our statement of poor study designs, we explain some of the shortcomings of previous literature in more detail, i.e. starting with line 58 we address the lack of holistic interventions, starting with line 81 we address the need of conceptualizing movement behavior as compositional and starting with line 96, we address the need of measuring physical activity measures using device-based methods rather than by self-reports.

L.98-99: for how long the interventions. The introduction did not highlight a research gap related to interventions duration.

Answer: Thank you for pointing this out. We have described the length of the intervention later in the text but we understand that the reader should get this information earlier on. We have now added this information in line 101:

 “Addressing these research gaps, the present study used data from a three-armed multi-component 6-months cluster randomized controlled trial (RCT) to (…)” (Line 101)

However, we did not identify intervention length as a research gap. It is indeed very interesting to discuss the issue of intervention length, but we decided to reflect upon it only shortly in the discussion section.

L131. Not knowing the mean age is hard to support the findings. How can age affect that? Also, there are no information regarding sample characteristics, such as BMI. This will difficult the results discussion.

Answer: Sample characteristic, including age, gender and duration of education, are provided in table 1. The mean age of the participants was 43 years. We have now added BMI and cardiorespiratory fitness to this table. Thank you for this suggestion.

Table 1. This reviewer believes that this information must be in methods regarding the sample characteristics.

Answer: In line with most other studies, we prefer to have table 1 as part of the results section. However, if the editor wants us to move it to the methods section, we can do that on eventual request.

This reviewer did not clearly understood the intervention. Was this multicomponent "training"? Exercise? Physical activity? What type of exercises?

 Answer: The interventions consisted of multiple components which addressed the individual participant as well as the organization and environment they were working in. The aim was to support the participants in changing their behavior.

We have made some changes to the section 2.2.Interventions starting in line 157 to describe the intervention more clearly:

Previous version: “The two multi-component interventions, iPA and iSED, were designed to influence behavior at multiple levels, i.e. the individual, the environmental and the organizational level, based on ecological models for health behavior [11,31] with the ultimate goal of improving mental health and cognition (See Figure 1). Both interventions lasted for six months and were similar in design, while focusing on different behaviors.”

New version: “The two interventions, iPA and iSED, included multiple components intended to influence behavior at different levels, i.e. the individual, the environmental and the organizational level, based on ecological models for health behavior [11,31] with the ultimate goal of improving mental health and cognition (see Figure 2). Participants decided together with coaches which type of activities would accommodate their needs and preferences in order to achieve a sustainable behavior change. Both interventions lasted for six months and were similar in design, while focusing on either increasing PA or reducing SED.”

We have also slightly changed the headers and footer in figure 2 to clarify that there are different intervention components, addressing different levels of behavior determinants.

In addition, we changed the wording in line 63-65 of the introduction:

Previous version: “Concerning the individual level, cognitive behavioral therapy (CBT) techniques in combination with motivational interviewing (MI) have been used widely to improve various health behaviors [14-17].”

New version: “Concerning the individual level, cognitive behavioral therapy (CBT) techniques in combination with motivational interviewing (MI) have been used widely to support people in changing various health behaviors [14-17].”

Discussion: Based on the interventions characteristics, the results are prone to be predicted. PA sessions are not training or exercise, some aspects such as load and individualization are not controlled. How can that explain the results? Please highlight.

 Answer: The aim of this study was to find out whether two multi-component interventions were successful at helping participants change their behavior either to integrate more physical activity in their lives or to reduce sedentary behavior. Of course, it is also interesting to find out how much load and which types of exercises are needed to achieve an improvement in mental health and cognition, as has been done in numerous exercise studies. However, this was not the aim of this study.

We have now added information to the introduction section to clarify that the aim of the individual component of the intervention was to providing participants with tools that can improve their intrinsic motivation and self-regulation:

“CBT strategies focus on providing people with concrete tools for achieving and sustaining behavior change by supporting their intrinsic motivation and self-regulation skills. Previous research has shown that people who are motivated by their own needs and desires, find it easier to sustain new behaviors.” (Lines 67-71)

In addition, we have changed the wording and added information to section 2.2 describing the interventions to clarify that it was up to the participants to which types of physical activity to integrate into their lives: 

“The two interventions, iPA and iSED, included multiple components intended to influence behavior at different levels, i.e. the individual, the environmental and the organizational level, based on ecological models for health behavior [11,28] with the ultimate goal of improving mental health and cognition (see Figure 1). Participants decided together with coaches which type of activities would accommodate their needs and preferences in order to achieve a sustainable behavior change. Both interventions lasted for six months and were similar in design, while focusing on either increasing PA or reducing SED. A team leader was appointed to each cluster and instructed to implement a set of intervention components as described below (Figure 1). Three counselling sessions were individual and two were group sessions.” Lines (157-166)

Thus, we were not interested in forcing or even prescribing subjects to perform a certain dose of a particular exercise or training type – as is done in exercise studies. We rather attempted to facilitate a behavior change by providing participants with the possibility to individually develop tools for behavior change (individual level) and by making changes in their environment and within the organization (environmental and organizational level).

We hope that the changes made in line with your previous comment (see above) clarify the intend of our interventions for the readers.

In the discussion section, we discuss how each of the intervention components might have impacted the results.

L404: should be ) instead of =

Answer: Thank you, this is now changed accordingly.

Round 2

Reviewer 1 Report

I thank the authors for their responses and adjustments.

There is still an issue that I want to raise:
The authors state that a complete cases study is more robust. Can they provide evidence for that statement?

I think that a complete cases analysis will cause a loss of power and a missed chance to compensate somewhat for missing conditionally at random. In particular while there is a selective drop-out (iSED, age, education).
The authors did not find a significant difference between the groups. It should be strong when they showed that they have paid every effort to find one. Regard the all-cases analysis as a sensitivity analysis. Please try it. Maybe it is not necessary to report it extensively.

Author Response

Response to reviewers, round 2

Reviewer 1:

There is still an issue that I want to raise:
The authors state that a complete cases study is more robust. Can they provide evidence for that statement?

I think that a complete cases analysis will cause a loss of power and a missed chance to compensate somewhat for missing conditionally at random. In particular while there is a selective drop-out (iSED, age, education).
The authors did not find a significant difference between the groups. It should be strong when they showed that they have paid every effort to find one. Regard the all-cases analysis as a sensitivity analysis. Please try it. Maybe it is not necessary to report it extensively.

Answer:

Thank you for being persistent in wanting us to explore all possible differences between the groups by performing an all-case analysis in addition to the complete case analysis.

As mentioned before, we decided for the complete case analysis also because the published effectiveness study (Nooijen et al. 2020) did not find an intervention effect, neither when complete cases were used, nor when all cases were analysed (intention to treat analysis).

However, we have now performed and added a sensitivity analysis including all cases, as you suggested.

Please see additions made to the statistical analysis section:

“An intention to treat analysis with all included participants (N = 263) participants was performed as a sensitivity analysis for all outcomes.” (Lines 294-295)

And the results section on movement behaviors: 

“The sensitivity analysis based on all included participants (N = 263) confirmed the lack of intervention effect.” (Lines 364-365)

And the results section on cardiorespiratory fitness: 

“The sensitivity analysis based on all included participants (N = 263) confirmed the lack of intervention effect.” (Lines 383-384)

Reviewer 2 Report

The authors have answered some of the questions. However, as far this reviewer understood, the authors do not even know if they applied a physical activity or exercise program.

They did not control intensity, effort or other variables related to the exercise’s intensity. They did not answer which type of exercises were used.

In fact, this reviewer identified that the authors cannot detail the intervention program characteristics because they were not controlled. Upon that, the results are too much subjective and dependent of participants individual characteristics.

Not providing sufficient information about the intervention program, such as load, exercises and intensities, may lead to type I or II errors.

Author Response

Response to Reviewer 2, round 2:

The authors have answered some of the questions. However, as far this reviewer understood, the authors do not even know if they applied a physical activity or exercise program.

They did not control intensity, effort or other variables related to the exercise’s intensity. They did not answer which type of exercises were used.

In fact, this reviewer identified that the authors cannot detail the intervention program characteristics because they were not controlled. Upon that, the results are too much subjective and dependent of participants individual characteristics.

Not providing sufficient information about the intervention program, such as load, exercises and intensities, may lead to type I or II errors.

Thank you for your comment. We would like to clarify again that this study was not designed to investigate physiological responses to predefined intensities of exercise. In the first sentence of our

manuscript, we cite the Physical Activity Guidelines Advisory Committee Scientific Report; Washington

DC, 2018 as one of many examples of as synthesis of such evidence. While these such investigations are

highly valuable for understanding what dose of exercise is needed to promote health, they often to not

address the question of how such behavioral change can be sustainably achieved.

With our behavior change study we are taking the next step, from proof of concept to translating evidence to practice. Knowing that physical activity has positive effects, we want to find out how we can help office workers to integrate more physical activity or less sedentary behavior in their lives. Therefore, we have designed two interventions which are described in detail in figure 2 as well as in section 2.2 Interventions. These interventions includes different components that address the individual level, but also make changes to the office workers’ environment and organization they are working in.

Thus, the aim of our study was to find out whether our interventions (which consist of multiple components, as detailed in fig. 2) can help people to change their movement behavior, i.e. increasing the time they spent in moderate to vigorous physical activity in one of the intervention groups and decreasing their time spent in sedentary behavior in the other intervention group.

In line with this, our outcomes were accelerometer-measured times spent in behaviors at different intensities.

In an attempt to make it very clear to all readers that our study is a behavior change study and not an exercise study, we have now made further adaptations to the manuscript:

  1. We added a sentence in the very beginning of the intervention to place the topic of behavior change as early on as possible:

“While this extensive research about the positive effects of a more active lifestyle on health outcomes exist, we need to understand more about how people can change their movement behavior.” (Lines 50-52)

  1. We made changes to following sentence (Lines 54-57):

Previous version: “Numerous studies have tried to find effective strategies for improving office workers’ movement behavior by either decreasing SED, increasing PA, or combining both of these, in order to improve various health outcomes [6-9].”

New version: “Numerous studies have tried to find effective strategies that help office workers to improve their movement behavior by either decreasing SED, increasing PA, or combining both of these, in order to improve various health outcomes [6-9].”

  1. We have moved the detailed description of the CBT- and MI-techniques - which were used to address the individual level of behavior - from the discussion section to the methods section. (Lines 165-170).

  1. We have added a reference to a sentence that we added in the last round of revision:

“Participants decided together with coaches which type of activities would accommodate their needs and preferences in order to achieve a sustainable behavior change.” (Reference: Samdal, G.B.; Eide, G.E.; Barth, T.; Williams, G.; Meland, E. Effective behaviour change techniques for physical activity and healthy eating in overweight and obese adults; systematic review and meta-regression analyses. Int J Behav Nutr Phys Act 2017, 14, 42, doi:10.1186/s12966-017-0494-y.) (Lines 163-165)

We highly recommend a read of this article to the reviewer as it presents a wide range of behavior change studies.

Thank you once again for providing us with an opportunity to further clarify the purpose of this study.